# Water Softening Using a Light-Responsive, Spiropyran-Modified Nanofiltration Membrane

**DOI:** 10.3390/polym11020344

**Published:** 2019-02-15

**Authors:** Rasel Das, Mathias Kuehnert, Asieh Sadat Kazemi, Yaser Abdi, Agnes Schulze

**Affiliations:** 1Leibniz Institute of Surface Engineering (IOM), Permoserstr, 15, 04318 Leipzig, Germany; raseldas@daad-alumni.de (R.D.); Mathias.Kuehnert@iom-leipzig.de (M.K.); 2Nanophysics Research Laboratory, Department of Physics, University of Tehran, Tehran P.O. Box 14395/547, Iran; asiehsadat.kazemi@ut.ac.ir (A.S.K.); y.abdi@ut.ac.ir (Y.A.)

**Keywords:** water softening, photo switching, salt rejection, polyamide, nanofiltration, electron beam irradiation

## Abstract

A novel technique for the covalent attachment of a light-responsive spiropyran onto polyamide thin film composite nanofiltration (NF) membranes in a one-step reaction using low-energy electron beam technology is described. The effect of illumination of the immobilized spiropyran was studied, as well as the resulting membrane properties with respect to MgSO_4_ retention, water permeability rate, and chlorine resistance. Electron beam irradiation showed a direct effect on the transformation of the rough PA NF membrane surface into a ridge-and-valley structure. Upon UV light irradiation, the spiropyran transformed into zwitterionic merocyanine, which had shown MgSO_4_ removal of >95% with water permeation rates of 6.5 L/(m^2^·h·bar). Alternatively, visible light was used to convert merocyanine to spiropyran, which achieved >95% of MgSO_4_ retention with a water flux of around 5.25 L/(m^2^·h·bar). The modified NF membranes showed higher chlorine resistance as well as a higher normalized water flux as compared to the reference membrane, without a loss of ion retention. All the NF membranes were characterized by scanning electron microscopy and X-ray photoelectron spectroscopy. This study demonstrates a simple and inexpensive method for the immobilization of molecules onto polymeric membranes, which may be applied in water softening.

## 1. Introduction

Hard water refers to a high mineral content, especially when loaded with large amounts of magnesium and calcium ions. Hard drinking water damages industrial settings, e.g., breaking boiler pipelines and other instruments that handle hard water. It thwarts the foam formation when water minerals are agitated with soap, which decreases cleaning efficiency. Moreover, it induces the formation of limescale in kettles and water heaters, and could damage dishwashers, washing machines, heat exchangers, and steam irons [1]. Hard water minerals make the water turbid and cloudy (milky), and the effect is especially evident in swimming pools. Furthermore, hard water causes the deposition of insoluble metals or salts in containers, sinks, and bathtubs [2]. Hence, removing these excessive mineral contents from water through a process called water softening is a must to avoid these problems.

Many methods have been developed for water softening, such as membrane technologies including nanofiltration (NF) and reverse osmosis (RO), electromembrane, e.g., electrodialysis, chemical precipitation, ion exchange process, etc. [3]. Due to their high stability, greater permeation flux and ion retention ability, process intensity, lower chemical consumption, automated process control, and operational robustness, membrane technologies have become one of the most widely used water softening methods in recent decades [4]. The pitfalls of the remaining aforementioned processes include the requirement of a large amount of chemicals, high power consumption, and high expenses for the maintenance and operation of the equipment [5].

Polyamide thin film composite (PA-TFC)-based NF membranes dominate the water softening debate [6] due to their universal application. However, optimization is still required. The majority of PA-TFC membranes that have been developed are static in both form and function. Therefore, designing stimuli-responsive dynamic NF membrane is innovative. Although there are different forms of external stimuli that can influence the behavior of a PA-TFC membrane, light has unique advantages. Light can be delivered in temporal precision, no chemical contaminants are introduced, a closed system can be actuated, and light of a specific wavelength can be delivered [7].

For the first time, we report here that electron beam (EB) technology can be efficiently used for the fabrication of a light-responsive PA-TFC-based NF membrane using photochromic spiropyran (SP) for water softening. EB activates the surface of PES/PA-TFC membrane and subsequent immobilization of SP using low-energy EB in a water/ethanol solution. The EB treatment results in the formation of different activated species such as radicals. These radicals can undergo recombination reactions to form covalent bonds [8,9,10]. This one-step method urges neither any preceding surface functionalization nor the use of additional chemicals or other toxic reagents. Therefore, EB provides a directed, fast, and environmentally friendly method for molecule coupling. The EB operates independently to functionalize any PA-TFC with specific desired functionalities. However, SP in its closed form is relative hydrophobic. Upon irradiation with ultraviolet (UV) light (323–351 nm) it switches to the relative hydrophilic merocyanine (MC) (opened form). In the presence of visible light (530–600 nm) the closed SP can be regenerated as shown in Figure 1. An interesting feature of using SP is its zwitterionic MC form (Figure 1). These features could be used to increase water permeability and rejection of MgSO_4_ ions.

## 2. Materials and Methods

### 2.1. Materials and Reagents

Polyethersulfone (PES) membranes (Express Plus, 0.22 µm) were purchased from Millipore (Merck Millipore, Billerica, MA, USA). Piperazine (PIP), trimesoyl chloride (TMC), *n*-hexane, ethanol, sodium hypochlorite, hydrochloric acid, and sodium hydroxide were purchased from Sigma-Aldrich (St. Louis, MO, USA). Hydroxylated SP was bought from Tokyo Chemical Industries Company Ltd. (Tokyo, Japan). Deionized water (pH 6.8) was used in all of the experiments.

### 2.2. Preparation of the PES/PA-TFC Membrane

A PES membrane (0.22 µM) was used as support for fabricating the PES/PA-TFC membrane. The PES was clamped into a handmade stainless steel frame as shown in Appendix A. Such a setup ensured that the interfacial polymerization (IP) reaction [11] could take place only on the top side of PES. Then the IP reaction took place by wetting the PES membrane surface with 1 wt % of PIP solution. After 10 min the PIP solution was drained off completely. The organic phase was prepared by dissolving 0.2 wt % TMC. It was subsequently introduced to cover the surface of the PES-PIP support for 2 min. After draining off the excess TMC solution, the PES/PA-TFC membrane was dried at 37 °C for 30 min. The resulting PES/PA-TFC membrane was washed with DI water for 30 min and stored in fresh DI water before further modification and/or use.

### 2.3. Functionalization of the PES/PA-TFC Membrane Using SP

A novel technique for the covalent immobilization of SP onto a PES/PA-TFC membrane was performed in a one-step reaction using low-energy electron beam. The method is illustrated in Figure 2.

The PES/PA-TFC membrane was placed on a glass plate. The modification was performed by immersing the PES/PA-TFC into an ethanol solution of SP at room temperature, followed by EB irradiation. The control experiment of PES/PA-TFC was performed by similar treatment without impregnation with SP. Concentrations of the SP were varied in the range of 0.02, 0.05, 0.10, 0.20, 0.40, and 0.80 wt % for impregnation of the membranes, and irradiation dose was in the range of 50–150 kGy. EB irradiation was performed in a N_2_ atmosphere with O_2_ quantities <10 ppm using a custom-made electron accelerator. The voltage and the current were set to 160 kV and 10 mA, respectively. The absorbed dose was adjusted by the speed of the sample transporter. The modified membrane was rinsed three times with ethanol and afterwards washed one time with DI water. The subsequent PES/PA-TFC and PES/PA-TFC-SP membranes were stored in DI water until further use.

### 2.4. Membrane Characterizations

The surface morphology of the reference (PES/PA-TFC) and sample (PES/PA-TFC-SP) membranes was investigated using an Ultra 55 scanning electron microscope (SEM, Carl Zeiss Ltd., Göttingen, Germany). The samples were manually cut and subsequently coated with a thin (30 nm) chromium film using the Z400 sputter system from Leybold (Hanau, Germany). 

The chemical composition was determined using X-ray photoelectron spectroscopy (XPS, Kratos Axis Ultra, Kratos Analytical Ltd., Manchester, UK). 

Fluorescence measurements of the samples were done using Infinite-M200, TECAN, Mannedorf, Switzerland. 

UV and visible light irradiations were carried out using CAMAG UV lamp (8 W, 366 nm) and LED (3 W, 530 nm) for 10 and 30 min, respectively. 

### 2.5. Chlorine Treatment

The PES/PA-TFC and PES/PA-TFC-SP membranes were immersed into a NaClO solution (2000 ppm). The chlorine exposure time was 1 h. The chlorination intensity is customarily measured with the product of chlorine concentration and exposure time in the unit of ppmh. The pH values of the solutions were adjusted to 7.0 with HCl or NaOH, respectively. Prior to a permeation experiment, the chlorine-treated membrane was rinsed with de-ionized water for 24 h to remove any residual chlorine in the membrane. Then, the water flux and salt retention were determined.

### 2.6. Membrane Performance Tests

The separation performance of the PES/PA-TFC and PES/PA-TFC-SP membranes was evaluated in terms of the water flux and salt rejection. Water permeability was determined using a custom-made cross flow system and a commercial cross flow cell (CF042D, active membrane area 42 cm², Sterlitech). The pressure was set to 15 bar and the cross flow was adjusted to 60 L/h. After compacting the membrane for 1 h the water permeability and MgSO_4_ (2000 ppm) retention were determined. Every membrane was tested three times. The permeation flux (*J*) was calculated using Equation (1): (1)J=Q/A·t,
where *Q* means the permeate volume (L), *A* depicts the PES/PA-TFC and PES/PA-TFC-SP membranes active area in m^2^ and *t* means time in h.

The concentration of MgSO_4_ in the test samples and the water permeate was calculated using a Mettler Toledo FE30 FiveEasy benchtop conductivity meter (Fisher Scientific GmbH, Schwerte, Germany). The percentage of MgSO_4_ rejection (*R*) by the PES/PA-TFC and PES/PA-TFC-SP membranes was calculated using Equation (2): (2)$$R(\%)=(1-C_{p}/C_{f})\cdot 100,$$
where *C*_f_ means the concentration of the feed solution and *C*_p_ is the concentration of permeate.

## 3. Results and Discussion

### 3.1. Membrane Characterization

#### 3.1.1. Morphology

The surface morphology (A), cross section (B), bottom side (C) and EB-irradiated PES/PA-TFC membrane surface (D) were visualized by SEM as shown in Figure 3. Figure 3A shows the overall view of the PA layer on the top of PES, and the high magnification image (a) reveals a PA layer of high roughness. The dense active layer of PA with a thickness of 50–70 nm on the porous PES is evident in Figure 3B. Figure 3C shows that all the pores at the bottom side of PES were open. The stainless steel frame used for the IP reaction (see Appendix A) could help protect the bottom side of PES from PA layer formation. An interesting ridge-and-valley surface morphology was found after EB irradiation of the PA layer (Figure 3D). It might be due to the EB-mediated radical formation, which causes internal polymer crosslinking, as we proved in our earlier study [12]. However, SEM images confirm that the presence of SP with different wt % inhibit the formation of this ridge-and-valley structure (see Appendix A). Thus, the SP-modified PES/PA-TFC-SP membrane (Figure 3E,F) showed a similar rough structure after EB modification as presented in (A). We assume that the EB-mediated covalent attachment of SP on top of the PA layer avoids additional crosslinking reactions within the PA polymer.

#### 3.1.2. Chemical Composition

PES/PA-TFC and PES/PA-TFC-SP membranes were analyzed using XPS. From the wide-scan spectra, O and N surface concentrations were determined as shown in Figure 4. It is visible that the O amount sharply increased with increasing SP concentration at SP 0.2 wt % and then a plateau is reached. It indicates considerable increased immobilization of SP at an optimum concentration of 0.2 wt % as most of the oxygen was sponsored by the hydroxylated SP. 

At the same time, increasing SP wt % resulted in a decreased N concentration. It might be due to the covalent attachment of SP with imide group of polyamide membrane. These can be well explained by Figure 5 that shows the corresponding C1s, O1s and N1s spectra. From the C1s spectrum depicted in Figure 5a, it can be seen that the four peaks can be assigned to the different types of C found in the backbone of PES/PA-TFC, including 285.00, 286.13, 287.25 and 289.15 eV due to the p–p* shake-up satellites, C–N, C=O and –COOH, respectively [13]. The C=C bond was contributed by the monomer TMC. Both the C=O and C-N groups were sponsored by amide bond of polyamide layer. The –COOH group was produced due to impartial IP reaction [14]. For the PES/PA-TFC-SP membrane, the intensity of C=C bond increased sharply due to the presence of SP. However, the broadened and intense peaks of PES/PA-TFC, i.e., C‒N and COOH decreased sharply in PES/PA-TFC-SP because of covalent attachment of SP to the imide bond and the free carboxylic group of the polyamide surface. Decreased peaks of N1s in PES/PA-TFC-SP as compared with PES/PA-TFC also supported this observation. However, the covalent bond could also be formed between the hydroxylated SP with the C=O group of PES/PA-TFC as the peak had vanished in PES/PA-TFC-SP membrane. Similar data gained by the O1s spectra support this assumption. The intensity of C=O group at 532.34 eV in PES/PA-TFC decreased in PES/PA-TFC-SP. At the same time, the intensity of C‒O peak at 533.14 eV increased in PES/PA-TFC-SP due to the presence of SP.

In order to further confirm the presence of SP on PES/PA-TFC, the photographic images of PES/PA-TFC and PES/PA-TFC-SP membranes could be considered as shown in Figure 6 (left). The white PES/PA-TFC-SP membrane became purple by isomerization of SP to MC using UV light (366 nm). After exposure to visible light, the SP form is regenerated and the membrane appears white again. Obviously, SP and MC differ markedly in their light absorption behavior. Therefore, light absorption measurements have been conducted (Figure 6 right). While the SP isomer exhibits strong absorption peak at λ_max_ ≈ 500 nm the MC form shows an intense absorption band centered at λ_max_ ≈ 650 nm. We also optimized the EB dosages (50 and 150 kGy) in order to check an appropriate EB energy for covalent immobilization of SP onto PA-TFC membrane. A very high absorption intensity of MC was recorded in case of a low EB dose of 50 kGy as compared to higher EB dose of 150 kGy upon SP irradiation at 366 nm. Therefore, 50 kGy of EB dose was taken into consideration for further experiments.

### 3.2. Separation Properties of the PES/PA-TFC-SP/MC Membranes

As described above, the water permeability and salt retention were investigated by cross-flow filtration and conductometry experiments, respectively (presented in Figure 7). To do so, membranes with the active SP or MC form present on the surface were investigated, respectively. To generate the MC form, PES/PA-TFC-SP membranes were exposed to UV light for 10 min (see Section 3.1.2 and Figure 6) before characterization. 

The reference membrane (PES/PA-TFC) exhibited water permeability around 6.3 L/(m^2^·h·bar). Upon EB modification of PES/PA-TFC, water permeability decreased to around 5.0 L/(m^2^·h·bar). Since EB activation causes cross-linking within the membrane polymer [10], it might shrink the pore diameter and make the membrane surface more dense than the reference PES/PA-TFC membrane. The SEM images of the PES/PA-TFC revealed a rough PA layer formation, whereas ridge-and-valley surface morphology was observed in the case of EB-irradiated PES/PA-TFC membrane (see Section 3.1.1). However, the water permeability of PES/PA-TFC-MC membranes was further reduced to around 4.7 L/(m^2^·h·bar) at very low as well as at the highest tested MC concentrations of 0.02 and 0.80 wt %, respectively. All other remaining PES/PA-TFC-MC membranes showed higher water permeability, they were even higher, i.e., 6.5 L/(m^2^·h·bar) for the 0.1 wt % MC compared to the reference membrane. Within this optimized MC concentration range, the hydrophilic character of the ionic MC (Figure 1) present on the membrane surface leads to an improved flux of the membrane. This is higher than the previously published NF membranes [15] NTR7450 [16] and commercial AFC40 [17]. These water fluxes were achieved without the loss of ion selectivity as all investigated membranes had a MgSO_4_ rejection >95%.

Similar to the discussion of MC-modified membranes above, the water flux and salt retention of PES/PA-TFC-SP membranes were characterized by cross-flow filtration and conductometry experiments, respectively. The lowest concentration of SP, i.e., 0.02 and 0.05 wt %, resulted in membranes with the highest water flux, i.e., just above 5.3 L/(m^2^·h·bar), but lower than the MC-modified membranes (except 0.02 wt % MC). All other remaining PES/PA-TFC-SP membranes showed lower water fluxes. This can be explained with the hydrophobic character of the nonionic SP (Figure 1) present on the membrane surface. The results clearly demonstrate that the EB immobilization of SP resulted in a light-responsive PES/PA-TFC membrane. The optimized concentration was determined to be 0.1 wt %. Thus, water flux was switched from 4.6 L/(m^2^·h·bar) to 6.5 L/(m^2^·h·bar) only by UV exposure for 10 min, which initiates the isomerization of hydrophobic nonionic SP to hydrophilic zwitterionic MC on top of the membrane.

### 3.3. Chlorine Treatment

Chlorine is commonly used as a water disinfectant. Severe chlorine treatment typically causes a performance decrease of PES/PA-TFC membranes in terms of water flux and salt rejection. This was explained with a chlorine attack followed by hydrolysis of the amide bonds (–CO–NH–) of the PA backbone [18,19]. We determined normalized water flux and salt rejection in order to investigate the effects of EB and SP/MC concentration on the chlorine resistance of PES/PA-TFC membranes as shown in Figure 8. The reference membrane experienced a water flux decrease to result in 58% after chlorine treatment. Regarding the EB treated membrane the water flux resulted in 66% indicating an improved stability by crosslinking. Modification with lower SP concentrations (i.e., 0.02 wt %) resulted in high water flux (90%). Membranes modified with 0.1, 0.2, and 0.4 wt % SP showed a resulting water flux comparable with the EB treated membrane. Interestingly, no loss of salt rejection was found regarding all studied membranes. These findings were well supported by XPS data (see materials Appendix A), where the highest amount of chlorine was detected in the reference membrane, whereas the EB-modified and SP-modified membranes possess lower chlorine amounts. This can be explained by the reaction process of PA chlorination. The degradation of PA membranes normally causes N chlorination of the amide nitrogen and ring chlorination [18,19]. N chlorination includes the substitution of hydrogen and attachment of chlorine onto the amide nitrogen to form N-chloroamide. Furthermore, an intramolecular Orton rearrangement is possible where the N-bonded chlorine atom can be removed to form molecular chlorine. This chlorine could attack the aromatic ring via electrophilic substitution. It yields an indirect ring chlorination. Possibly, SP was covalently linked to this vulnerable amide nitrogen N, so that N-chloroamide formation and ring chlorination in PA-TFC-SP might be prohibited.

## 4. Conclusions

Light-responsive spiropyran immobilization onto PES/PA-TFC membranes was successfully accomplished using EB technology. SEM images demonstrated that the EB-irradiated membranes possess a ridge-and-valley structured PA surface compared to a rough structured reference surface. These ridge-and-valley PES/PA-TFC membranes showed a drop in water permeability rate while MgSO_4_ rejection remained >95%. However, EB technology can retain the rough structured PES/PA-TFC in the presence of spiropyran. UV light irradiation was used to convert spiropyran into the zwitterionic merocyanine. It ensured MgSO_4_ ions removal >95% with water permeation rates of 6.5 L/(m^2^·h·bar). Alternatively, merocyanine was switched back to spiropyran upon visible light irradiation that confirmed >95% of MgSO_4_ retention with water flux around 5.25 L/(m^2^·h·bar). The spiropyran immobilized PES/PA-TFC-based NF membranes had a higher chlorine resistance and showed higher normalized water flux as compared to the reference membrane without the loss of ion retention.

## Figures and Tables

**Figure 1 polymers-11-00344-f001:**
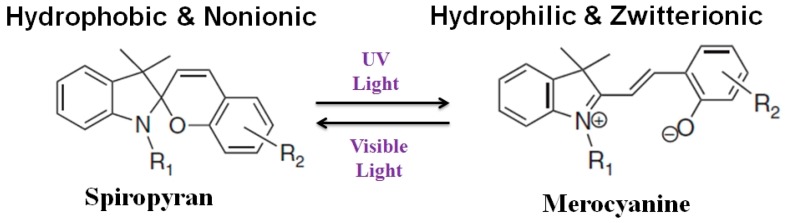
Photo switching of spiropyran.

**Figure 2 polymers-11-00344-f002:**
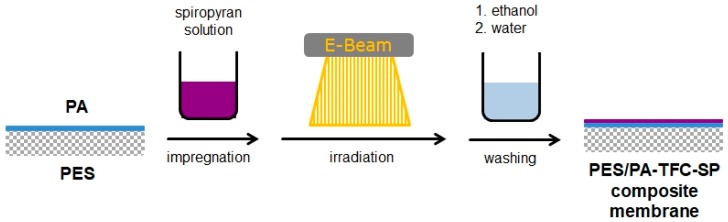
Illustration of the attachment of SP onto a PES/PA-TFC membrane surface using EB irradiation.

**Figure 3 polymers-11-00344-f003:**
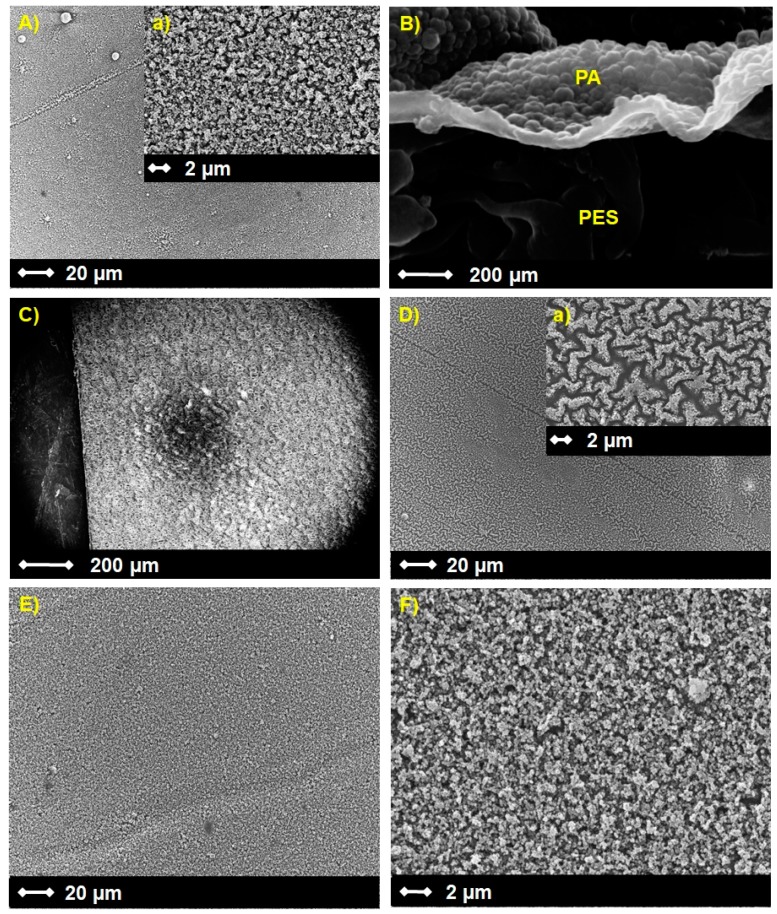
SEM images of surface (**A**), cross section (**B**), bottom side (**C**) and EB-irradiated PES/PA-TFC surface (**D**) as well as PES/PA-TFC-SP (0.1 wt %) surface in different resolutions (**E**,**F**). A and D have high-resolution SEM images insets (**a**).

**Figure 4 polymers-11-00344-f004:**
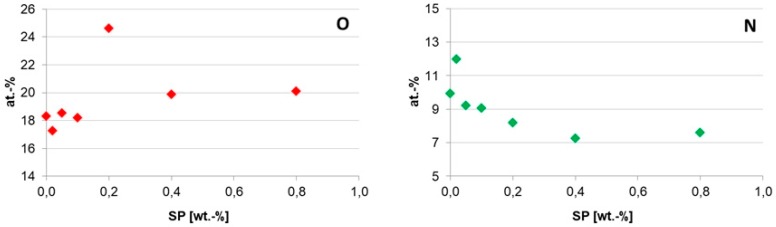
Elemental surface concentrations of O and N of the PES/PA-TFC and PES/PA-TFC-SP membranes as determined by XPS.

**Figure 5 polymers-11-00344-f005:**
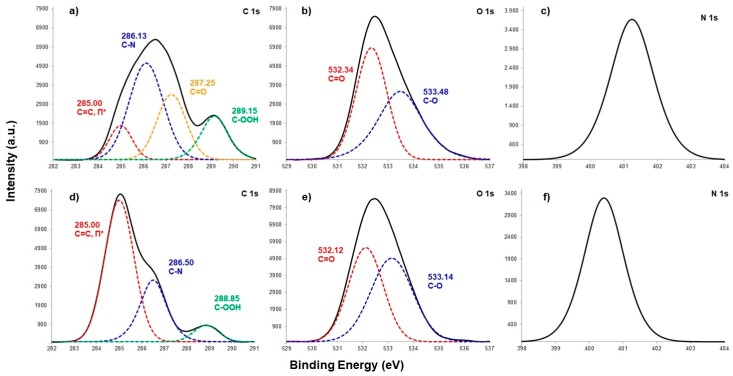
C1s (**a**,**d**), O1s (**b**,**e**) and N1s (**c**,**f**) spectra of PES/PA-TFC (**a**–**c**) and PES/PA-TFC-SP (0.2 wt %) (**d**–**f**) as determined by XPS.

**Figure 6 polymers-11-00344-f006:**
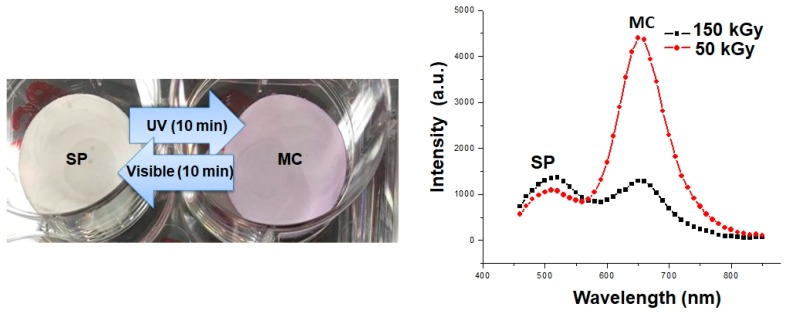
**Left**: photo of PES/PA-TFC-SP membrane after illumination with UV and visible light, respectively; **right**: absorption spectra of PES/PA-TFC-SP after irradiation with different electron beam doses.

**Figure 7 polymers-11-00344-f007:**
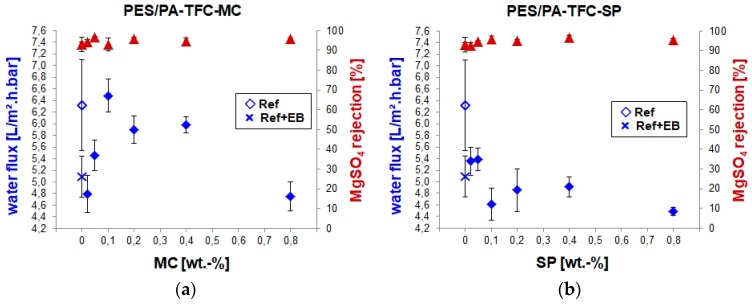
Water flux and MgSO_4_ retention of the PES/PA-TFC-MC (**a**) and PES/PA-TFC-SP (**b**) membranes.

**Figure 8 polymers-11-00344-f008:**
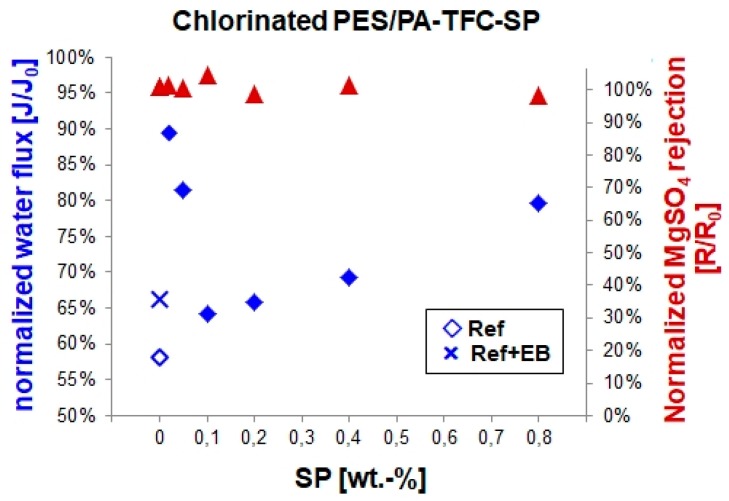
Normalized water flux and MgSO_4_ retention of the reference, EB-modified and PES/PA-TFC-SP membranes after chlorination.

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
