# Peer review of "Water Softening Using a Light-Responsive, Spiropyran-Modified Nanofiltration Membrane"

_polymers, 2019, doi:10.3390/polym11020344_

Round 1
Reviewer 1 Report
The authors carefully consider the comments. The manuscript is suitable for publication now.
Author Response
Many thanks to the reviewer. We appreciate the helpful comments that enabled us to improve the quality of our manuscript leading to the final conclusion of the reviewer that is suitable for publication now!
Reviewer 2 Report
The manuscript describes a modification of PA NF membranes through a layer of light reactive material. Although the approach is novel and interesting, the practical application of light in an NF membrane module is very limited. NF modules are arranged as spiral-wound with very limited space. Adding a light source to the high-pressure module will be expensive and simple non-practical.
In addition, the following suggestions are presented:
1. Line 80-81: “All the PA-TFC membranes that have been developed until this time are static in both form and function”. This statement is not true as there have been publications or responsive membranes which are impacted by pH, electrical potential and temperature.
2. Line 195 - cross flow is presented as 60 L/h. The true value should be presented as linear flow velocity which impacts the shear and concentration polarization (CP).
3. Line 339 – Defining roughness by SEM is very tricky, to really understand this value you need to add AFM results for each membrane structure. Also to understand water permeability through hydrophilicity and surface energy you should at least measure contact angle.
4. Section 385 – chlorine treatment:
5. Is the additional layer physically protecting the PA layer? What happens to it at increasing concentrations (time*concentration)? Please address this as well.
6. As for the flux values, flux could be impacted significantly by CP, any change in surface structure will impact the mixing near the surface and change the CP value. I think this should be also addressed.
Author Response
We appreciate the thorough examination of the manuscript by the referee! We carefully revised the manuscript according to the requested comments and questions. Therefore, we would like to resubmit our manuscript and hope we were able eliminate all mistakes and misunderstandings according to your suggestions.
Please find below our answers to the reviewer’s comments in red color.
The manuscript describes a modification of PA NF membranes through a layer of light reactive material. Although the approach is novel and interesting, the practical application of light in an NF membrane module is very limited. NF modules are arranged as spiral-wound with very limited space. Adding a light source to the high-pressure module will be expensive and simple non-practical.
We agree that the technical implementation of light sources into a spiral-wound module might be a challenge that still needs to be addressed. However, using LED strips or optical fibers could solve the mentioned space problems. This needs to be considered when the system should be transferred into a real application.
In addition, the following suggestions are presented:
1. Line 80-81: “All the PA-TFC membranes that have been developed until this time are static in both form and function”. This statement is not true as there have been publications or responsive membranes which are impacted by pH, electrical potential and temperature.
We rephrased the sentence:” The majority of PA-TFC membranes that have been developed until this time are static in both form and function.”
2. Line 195 - cross flow is presented as 60 L/h. The true value should be presented as linear flow velocity which impacts the shear and concentration polarization (CP).
As we intended to investigate the nature of surface chemistry by attaching single molecules on top of the PA membrane, we didn’t change the morphology of the membrane. Therefore, we believe that the comparison of water flux as presented in the manuscript (and as it is shown in the literature for comparable studies) is sufficient.
3. Line 339 – Defining roughness by SEM is very tricky, to really understand this value you need to add AFM results for each membrane structure. Also to understand water permeability through hydrophilicity and surface energy you should at least measure contact angle.
We agree, to quantify roughness and hydrophilicity, AFM and contact angle measurements would be necessary, respectively. However, we didn’t consider the value of roughness within our discussion or explanation. We just described the top surface of the membrane by SEM to qualitatively describe the structure changes caused by E-Beam treatment. This was a simple observation without any further conclusion. Therefore, we think that additional AFM measurements are not required. In further investigations, we will include also AFM to quantify the roughness effect if we believe that it is necessary to analyze the membrane properties.
Furthermore, we didn’t analyze the contact angle because we don’t have a setup that allows the contact angle determination during a selective light irradiation. This would be necessary to keep the surface in the SP or MC form, respectively. Since contact angle determination is usually performed by taking a photo from the drop of water that is placed on the surface against a light source, it was not possible to avoid switching back of the surface in our measurement setup. For the hydrophilic form of the membrane, however, we detected contact angles in the range of 22-24 °. This indicates a quite hydrophilic behavior of the membrane surface. For future experiments we plan to modify our test setup to enable the time-resolved analysis of the contact angle during light-selective irradiation.
4. Section 385 – chlorine treatment:
No comment/question that we could consider…
5. Is the additional layer physically protecting the PA layer? What happens to it at increasing concentrations (time*concentration)? Please address this as well.
The immobilized spiropyran molecule is attached as a single molecule layer on top of the PA membrane. Therefore, the physical nature of the membrane is not altered. Only the chemical properties are changed significantly by this modification. The immobilization is understood to be covalent and permanent. In previous studies, intensive extraction experiments revealed that the modification is not lost upon time, concentration and pressure treatments. We have shown and discussed this effect of single molecule immobilization by E-Beam already earlier in the cited papers [8]-[10]:
8. Schulze, A.; Marquardt, B.; Kaczmarek, S.; Schubert, R.; Prager, A.; Buchmeiser, M.R.,. Electron beam-based functionalization of poly(ethersulfone) membranes. Macromol. Rapid Commun. 2010, 31, 467-472.
9. Schulze, A.; Marquardt, B.; Went, M.; Prager, A.; Buchmeiser, M.R.,. Electron beam-based functionalization of polymer membranes. Water Sci. Technol. 2012, 65, 574-580.
10. Schulze, A.; Maitz, M.F.; Zimmermann, R.; Marquardt, B.; Fischer, M.; Werner, C.; Went, M.; Thomas, I.,. Permanent surface modification by electron-beam-induced grafting of hydrophilic polymers to pvdf membranes. RSC Adv. 2013, 3, 22518-22526.
6. As for the flux values, flux could be impacted significantly by CP, any change in surface structure will impact the mixing near the surface and change the CP value. I think this should be also addressed.
We agree. However, as commented also in point 5 the physical surface structure is not changed by the modification, therefore, we assume that flux is not effect by topography/morphology effects but by the surface hydrophilicity that changed.